

# Modeled changes in 100 year Flood Risk and Asset Damages within Mapped Floodplains of the Contiguous United States

Cameron Wobus[1], Ethan Gutmann[2], Russell Jones[1], Matthew Rissing[1], Naoki Mizukami[2], Mark Lorie[1], Hardee Mahoney[1], Andrew W. Wood[2], David Mills[1], Jeremy Martinich[3]

[1]Abt Associates, 1881 Ninth Street, Suite 201, Boulder, CO, 80302 USA
[2]National Center for Atmospheric Research, 3450 Mitchell Lane, Boulder, CO, 80301 USA
[3]U.S. Environmental Protection Agency, Climate Change Division, 1200 Pennsylvania Ave NW, Washington, DC, 20460 USA

*Correspondence to*: Cameron Wobus (Cameron_Wobus@abtassoc.com)

**Abstract.** A growing body of recent work suggests that the extreme weather events that drive inland flooding are likely to increase in frequency and magnitude in a warming climate, thus increasing flooding damages in the future. We use hydrologic projections based on the Coupled Model Intercomparison Project Phase 5 (CMIP5) to estimate changes in the frequency of modeled 1% annual exceedance probability flood events at 57,116 locations across the contiguous United States (CONUS). We link these flood projections to a database of assets within mapped flood hazard zones to model changes in inland flooding

damages throughout the CONUS over the remainder of the 21st century. Our model generates early 21st century flood damages that reasonably approximate the range of historical observations, and trajectories of future damages that vary substantially depending on the greenhouse gas (GHG) emissions pathway. The difference in modeled flood damages between higher and lower emissions pathways approaches $4 billion per year by 2100 (in undiscounted 2014 dollars), suggesting that aggressive GHG emissions reductions could generate significant monetary benefits over the long-term in terms of reduced flood risk.

Although the downscaled hydrologic data we used have been applied to flood impacts studies elsewhere, this research expands on earlier work to quantify changes in flood risk by linking future flood exposure to assets and damages at a national scale. This work uses relatively conservative assumptions and methods that ultimately affect damage estimates; future work is needed to test sensitivity related to these methodological choices (e.g., more sophisticated downscaling methods, use of multiple hydrologic models, and consideration of a wider range of flood magnitudes).

## 1 Introduction


Inland floods are among the most costly natural disasters in the United States (e.g., Pielke and Downton, 2000), with average annual damages ranging from hundreds of millions to many tens of billions of dollars over the past century (Downton et al., 2005; NOAA, 2016). In 2016, inland flooding events in Louisiana and North Carolina alone caused over $10 billion of damages to homes, businesses, and other assets (Fortune, 2016; LED, 2016). This follows on other recent years with historical flooding

in Michigan (2014) and Colorado (2013), and the mid-Atlantic floods caused by Superstorm Sandy in 2012 (NOAA, 2016).





With each occurrence of these damaging flood events, there is renewed interest in determining whether climate change may be partially responsible for changes in the magnitude or frequency of these events (e.g., IPCC, 2012; Trenberth et al., 2015). While the capacity to attribute an individual flood event to climate change remains challenging, the science linking changes in climate extremes to human-caused warming is advancing (e.g., Trenberth et al., 2015; National Academies of Sciences,

Engineering, and Medicine, 2016). In any case, projection of long-term trends in the frequency and magnitude of inland flooding remains important for municipalities, utilities, and national decision-makers as they assess their future infrastructure needs and climate vulnerabilities.

This study analyzes 21st century flood risk across the contiguous United States (CONUS) using downscaled hydrologic projections from 29 global climate models (GCMs) and 2 representative concentration pathways (RCPs) for greenhouse gas

(GHG) forcing extracted from the CMIP5 archive. We cross-referenced spatially explicit hydrologic projections with a database that catalogs assets and models flood exposure and damages within each of the mapped 1% annual exceedance probability (AEP; "100-year") floodplains in the CONUS. Using this combined dataset, we generate regional estimates of how modeled damages from what are currently 1% AEP events might change through the 21st century, and how these damages might differ under a higher GHG emissions scenario (RCP8.5) vs a lower emissions scenario (RCP4.5).

Our analysis builds on a deep body of previous work that has mined both historical hydrologic records and future climate projections to evaluate where detectable changes in extreme precipitation or flooding might have already occurred, as well as where we might expect these changes to occur in the future. For example, a number of studies have found statistically significant trends in streamflow for some regions of the United States (e.g., Lins and Slack, 1999; Mallakpour and Villarini, 2015; McCabe and Wolock, 2002). However, even where trends exist in these records, these trends are not always spatially

coherent (e.g., Archfield et al., 2016), and the extent to which these trends can be attributed to anthropogenic forcing is not always clear (e.g., Hirsch and Ryberg, 2012).

Because available hydrologic records tend to be short relative to the return interval of extreme flood events, analyses of historical flood trends are commonly inconclusive. Thus, there has been significant interest in using climate model outputs to evaluate future flood risk. Unfortunately, this is expensive computationally: at present the most widely used strategy for

assessing changes in future flood risk requires downscaling GCM outputs to hydrologically relevant spatial scales; estimating precipitation, infiltration, and runoff within a hydrologic modeling framework; and routing the resulting flows through a model river network (e.g., Das et al., 2013; Hirabayashi et al., 2013; Reclamation, 2014). Although less computationally demanding, studies attempting to link projected changes in extreme precipitation directly to changes in flooding (i.e., without a spatially explicit hydrologic model) tend to have high uncertainties (e.g., Kundzewicz et al., 2014; Wobus et al., 2014).

More recently, computational power has increased to the point that methods to downscale and route GCM-derived precipitation have become more readily available (e.g., Gosling et al., 2010; Hirabayashi et al., 2008; Reclamation, 2014). These outputs have been used to project future flood risk at scales ranging from local (e.g., Das et al., 2013) to global (e.g., Arnell and Gosling, 2016; Hirabayashi et al., 2013). However, to our knowledge there has not yet been a CONUS-scale assessment of how these changing flood risks could translate into monetary damages.



## 2 Methods

We used simulated daily hydrographs at 57,116 locations across the CONUS between 2000 and 2100 to calculate a CMIP5 modeled baseline ('current climate') 1% AEP event, and changes in the frequency of flows exceeding this magnitude through the 21st century. We quantified asset exposure and expected flood damage within mapped floodplains using a combination of

Federal Emergency Management Agency (FEMA) flood maps, US Census block data, and land cover data. Because only the "100-year" floodplains are consistently mapped and available at a national scale, our model of flood damages is driven only by changes in the frequency of what are currently 1% AEP events through the 21st century. We also do not project changes in floodplain development or flood protection through time, as such projections would require assumptions that are more difficult to justify at a national scale than for more regional or localized studies (e.g., Elmer et al., 2012). Our model projections should

therefore be considered order-of-magnitude estimates of how differences in emissions scenarios might propagate into changes in flood damages throughout the United States, based on available data from CMIP5.

### 2.1 Hydrologic Modeling Inputs

We used spatially and temporally disaggregated precipitation and temperature at 1/8th degree resolution from 29 GCMs and

2 emissions scenarios (RCP4.5 and RCP8.5), generated using the bias correction and spatial disaggregation (BCSD) method (e.g., Wood et al., 2004). The BCSD method uses a quantile mapping approach to match the distribution of GCM-derived monthly outputs to the observed monthly data at a 1-degree resolution in a historical period (1950–2000). It then uses the spatial pattern of daily observations from an analog month as a proxy for sub-grid scale daily (temporal) variability, and scales or shifts these daily observations to ensure that the analog monthly average values match the rescaled GCM output. During

the bias correction process (which applies to monthly precipitation and temperature values at the GCM scale), projected precipitation values exceeding the upper end of the climatological range are extrapolated following an extreme value Type I distribution. Additional details of the BCSD weather generation are given in Harding et al. (2012) and Wood and Mizukami (2014).

Catchment hydrology was simulated using the variable infiltration capacity (VIC) hydrologic model (Liang et al., 1994) forced

by the BCSD precipitation and temperature fields. The VIC model simulates the range of hydrologic processes relevant to generating runoff, including interception on the forest canopy, evapotranspiration, water storage and melt from snowpack, infiltration, and direct runoff. The runoff component of each model grid cell was remapped to the Hydrologic Response Units (HRUs) defined in the United States Geological Survey (USGS) Geospatial Fabric (GF; Viger and Block, 2014), and then routed through the GF river network using the MizuRoute routing tool, which incorporates both hillslope and river channel

processes (Mizukami et al., 2016a). The GF dataset contains ~57,000 river segments and ~108,000 HRUs (including the right and left bank of most river segments), representing catchments approximately equivalent in area to 12-digit Hydrologic Unit Code basins. The methods used for the downscaling and land surface hydrology were identical to those used in previous studies





(e.g., Das et al., 2013; Reclamation, 2014). However, for this effort we used a multi-scale parameter regionalization approach (Samaniego et al., 2010) to improve the spatial coherence of VIC model parameters across basin boundaries (Mizukami et al., In Review). Full details of the downscaling, VIC model parameters, and routing methodologies are available in Mizukami et al. (In Review) and Reclamation (2014).

Future changes in extreme precipitation are uncertain, but the methods used here are conservative with respect to predicting increases in precipitation. A growing body of work indicates that future extreme events are likely to increase more than the mean increase in precipitation (e.g., Kendon et al., 2014; Prein et al., 2016); however, BCSD only scales the extreme events with the mean changes. Although downscaling is required to simulate catchment hydrology at a physically meaningful scale, and the BCSD method has been used in the past to account for precipitation changes in hydrologic modeling applications (e.g.,

Das et al., 2013; Shrestha et al., 2014; Ning et al., 2015), downscaling methods are themselves imperfect. While BCSD has been shown to have fewer artifacts in historical climate compared to other commonly used methods (e.g., Gutmann et al., 2014), we show here that BCSD does introduce an artifact into the precipitation time series between historical and future projections. In particular, we found that the bias correction process generates a step change in the distribution of extreme precipitation events in the year 2000 (the break between the hindcast and the forecast periods in this BCSD application). As

summarized below, we accommodated this in our analysis by using an early 21st century ensemble average to represent baseline hydrologic conditions instead of the more traditional late 20th century. This choice is also conservative, as a longer time period starting in the late 20th century would increase the modeled changes, even if there were a way to do so without folding in the artifact introduced by the BCSD method.

## 2.2 Modeling Flood Probability

For each of the 58 GCM/RCP combinations in the hydrologic model output, we extracted the time series of annual maximum flow between 1950 and 2099 at each of the ~ 57,000 GF stream locations in CONUS. Average annual maximum flows in the modeled reaches range from $< 5$ m$^3$/s to $> 1,000$ m$^3$/s (Figure 1). Prior to generating statistics of peak flows from these events, we plotted the normalized annual maximum time series across all segments and all models (Figure 2). This plot revealed a step in the annual maximum flow time series in the year 2000, which corresponds to the end of the hindcast period used in the

BCSD method. This step is even more pronounced in the BCSD precipitation inputs (Figure 3), and most likely reflects the change in how the BCSD method constrains the distribution of events in the historical period compared to in the future period. In order to prevent this artifact from influencing our analysis of future flooding events, we calculated the magnitude of the "baseline" modeled 1% AEP flood event at each stream segment by fitting a generalized extreme value (GEV) distribution to the full ensemble of annual maximum flow estimates for each RCP over the 2001–2020 period (29 models x 20 years = 580

values), and extracting the 99th percentile value from this model fit. Although the emissions pathways for RCP4.5 and RCP8.5 begin to diverge in 2006, there were no systematic differences between GEV fits for the two RCPs, justifying our treatment of this early 21st century period as a baseline across the full ensemble.





Individual GCMs exhibit a degree of dependence due to shared code, shared scientific literature, shared observations, etc., and as such are not statistically independent (Abramowitz, 2010; Knutti et al., 2010b; Bishop and Abramowitz, 2013). However, the consensus of the community remains that it is best to average across many ensemble members (Tebaldi and Knutti, 2007; Knutti et al., 2010a) as we have done here. In addition, there were no systematic differences in results in the annual maximum

time series from the 29 individual GCMs, justifying our treatment of the full ensemble 580 of annual maxima when assessing peak flow magnitudes. From this full ensemble, we evaluated uncertainty in the 1% annual probability event by bootstrapping (see Supplemental Information File #1). Based on these analyses, we expect the sample uncertainty on our 1% AEP flood event to be in the range of 5–20%. As shown later, the variability in the multimodel GCM ensemble is much larger than this uncertainty in the GEV fits, so we did not propagate this source of uncertainty through all of our calculations.

To estimate future flood frequency and damages through the 21st century, we compared the full transient of future annual streamflow maxima for each GCM/RCP combination to the baseline 1% AEP event. In all of the summaries that follow, we define a "flood" at a given stream segment as an annual maximum flow value that exceeds the baseline 1% AEP event at that segment. The comparison between future flows and the 1% AEP threshold yields a time series of floods at each segment, as well as an estimate of the total number of flood events nationwide in each year. At each segment, we also calculated an

ensemble average probability of exceeding the 1% AEP event in each year, by tabulating the fraction of models experiencing a flood and smoothing these probabilities over a 20-year moving window. These time- and ensemble-averaged flood probabilities by segment were then linked to the assets exposed within each floodplain to calculate projected annual damages, as summarized below.

### 2.3 Asset Exposure and Damages

We estimated asset damages resulting from 1% AEP flood events using data from an experimental tool under development for the U.S. Army Corps of Engineers (USACE). For this tool, we compiled all of the 1% AEP floodplains as mapped by FEMA and included in the National Flood Hazard Layer (NFHL). We then used a series of steps to calculate the depth of flooding and resulting damages from 1% AEP events, and merged this information with the flood probabilities described in Section 2.2. A brief summary of the flood damage calculations follows. Supplemental Information File #2 provides more complete details

of the flood damage calculations.

### 2.3.1 Cataloguing Damages by Flood Zone

To catalog damages by flood zone, we intersected 1% AEP flood boundaries with Census blocks to create a set of flood zone polygons subdivided by Census block boundaries. Within each of these flood zone/Census block units, we created a random sample of flood depths for the 1% AEP event using the National Elevation Dataset (NED; USGS, 2016). We then merged this

information with land cover data from the National Land Cover Dataset (Homer et al., 2015) to determine the distribution of flood depths within "developed" and "undeveloped" portions of each Census block. For portions designated as "developed," we used the distribution of depths to estimate exposure of built assets using FEMA's HAZUS-MH General Building Stock





inventory (FEMA, 2009). The General Building Stock inventory provides estimates of the number and aggregate dollar value of multiple types of residential, commercial, and industrial buildings for each Census block.

For the developed portion of each Census block/floodzone intersection, we created damage estimates using depth-damage functions from USACE and FEMA (FEMA, 2009; USACE, 2000, 2003). A separate depth-damage function was used for each of 28 different categories of buildings (e.g., residential one-story homes without a basement). Each depth-damage function describes the percent loss as a function of depth. The depth-damage functions were applied to the aggregate value for each building category within each NFHL-Census block intersection, using the depth exposure results described above.

### 2.3.2 Aggregating Damages to National Scale

Once the damage estimates were generated for each Census block/floodplain intersection, we aggregated this information up to the same HRUs that were used in the hydrologic analysis. We then linked each stream segment at which flood statistics were calculated back to the total asset damages resulting from a 1% AEP event at that location. Figure 4 shows the total damages expected from 1% AEP events at each of the HRUs across the CONUS.

For each GCM, we combined the timeseries of floods at each stream segment with the assets exposed in that HRU to compute a timeseries of monetary damages. When averaged across all nodes in the CONUS, this approach yielded a relatively smooth curve of CONUS-wide monetary damages through the 21$^{st}$ century. However, this approach treats the hydrologic timeseries from each GCM as a deterministic, rather than a probabilistic, projection of future conditions. In order to use the full ensemble of GCMs in a more probabilistic framework, we used a Monte Carlo approach. We simulated 1,000 100-year time series of flood damages in the CONUS using the ensemble average probability of exceeding the 1% AEP event at each segment in each year. This yielded a distribution of flood damages in each year, from which we extracted a minimum, maximum, and ensemble average for each of the RCPs.

## 3 Results

Although each GCM/RCP combination yields its own time series of flooding, we have no a priori reason to focus on or exclude any individual models. Thus we center our discussion on the distribution of results across all 29 GCMs for each RCP.

### 3.1 Flood Frequency Projections

Since the hydrographs generated by the downscaled hydrology outputs are unique to each GCM/RCP combination, each model also produces its own time series of flooding at each stream segment. As one way of summarizing these data, we calculated the total number of flood events across the CONUS in each year of each model simulation. We then summarized the distribution of the total number of flood events across all 29 GCMs for each RCP (Figure 5). As expected based on our method, the annual number of 1% AEP floods across the CONUS across all models averages approximately 500 events between 2000



and 2020 (~1% of the ~ 57,000 segments in the CONUS). This average number of floods increases slightly to approximately 750 events by 2100 under RCP4.5, and up to approximately 1,250 events under RCP8.5.

Using the time series of flooding for each segment and combining these values across all models, we calculated an average flood frequency by segment for 20-year intervals in the baseline (2001–2020), mid-century (2040–2059), and late century (2080–2099). This allowed us to calculate an ensemble-averaged change in flood frequency for each segment, to evaluate where there may be spatially coherent patterns of increased flood risk. As shown in Figure 6, the largest fractional changes in flood frequency across the CONUS occur in the southern Appalachians and Ohio River valley, the northern and central Rocky Mountains, and the Northwest. In each of these regions, the ensemble average across models suggests that historical 1% AEP events could become 2–5x more frequent by the end of the century.

In some regions of the United States (e.g., the southern Appalachians and northern Rocky Mountains), the spatial patterns of increased flood frequency can be explained by the increased occurrence of extreme precipitation events projected by BCSD precipitation outputs. In other regions such as the Sierras and the Cascades, increases in the frequency of flood events are not as easily explained by changes in precipitation alone. In these locations, the increase in frequency of extreme floods more likely reflects changes in the nature of winter precipitation (rain vs snow) compared to baseline conditions (e.g., Das et al., 2013).

## 3.2 Flood Damage Projections

By combining the changes in frequency of flooding at each segment with the asset exposure and damage associated with each floodplain, we generated a full time series of projected changes in flood damages across the CONUS through the 21st century. Figure 7 shows the results from 1,000 individual simulations of nationwide flood damages using the probability of flooding at each segment, as described in Section 2.3.2. As shown in Figure 7, changes in flood damages broadly mimic changes in flood frequency at a national scale (Figure 5), with minor differences between these trends reflecting the way that regional trends in flood frequency interact with asset exposure within 1% AEP floodplains (see Figure 4). In particular, expected annual flood damages under the RCP4.5 scenario increase modestly from approximately $3 billion between 2000 and 2020 to approximately $4 billion by the end of the century. Under the RCP8.5 scenario, expected annual flood damages increase from approximately $3 billion in the early 21st century to over $7 billion by 2100.

Figure 7 also highlights how different GHG emissions pathways generate different trajectories of flood damages through the remainder of the 21st century. While the RCP4.5 and RCP8.5 pathways are generally similar through mid-century, the damage trajectories under the two emissions scenarios begin to diverge in the latter half of the 21st century. By 2075, the average annual difference between flood damages under the RCP4.5 and RCP8.5 emissions pathways is approximately $2 billion, and by 2100 this difference grows to almost $4 billion.

The increasing flood damages under RCP8.5 relative to RCP4.5 are not evenly distributed throughout the United States. Figure 8 shows the time series of average annual damages in each region of the CONUS. As shown in Figure 8, the most significant differences between projected flood damages under the two emissions scenarios are in the Southeast, where the difference





between the two trajectories approaches $2 billion per year by the end of the century; and in the Northeast, where the difference between the two scenarios is almost $1 billion per year by 2100. Although there are subtle differences in damage between the two emissions scenarios in other regions (e.g., the Southwest and the Midwest), these differences are small relative to those two regions of the country.

5 **4 Discussion and Conclusions**

Based on our model, we find that if future GHG emissions remain unchecked, monetary damages from flooding throughout the CONUS are likely to increase through the 21st century. Global GHG reductions, represented by RCP4.5, could limit these increasing flood damages, potentially saving up to $4 billion per year (in undiscounted 2014 dollars) by the end of the century. To our knowledge, this study is the first to link spatially explicit hydrologic projections from a full ensemble of climate model 10 projections to mapped assets in order to calculate future flood damages at a national scale.

Although this study represents a significant methodological advance in projecting future inland flooding damages nationwide, there are a number of caveats that must be noted. Perhaps most importantly, GCMs do not resolve precipitation well (e.g., Flato et al., 2013). The BCSD downscaling method used here was designed in part to improve the representation of historical precipitation, but our analysis shows that this method also introduces artifacts into the time series of extreme precipitation 15 between the historical and future projections that are not well-understood (e.g., Figure 3). The gradual increase in future precipitation extremes observed between 2000 and 2100 reflects increasing precipitation projected by the raw GCMs, which ultimately drives increases in the frequency of damaging floods in our modeling. However, an improved representation of precipitation extremes would improve confidence in our results.

Preliminary analysis of precipitation outputs from the newer localized constructed analogue (LOCA: Pierce et al., 2014) 20 statistical downscaling method suggests that artifacts introduced by the BCSD method are likely to exist in other products as well. Ideally, future hydrologic projections could be driven by a dynamically downscaled climate model to avoid these statistical artifacts; however, full dynamical downscaling through the 21st century at the scale of the CONUS may remain computationally prohibitive for a number of years to come (see for example Liu et al., 2016). In the interim, future work could replicate the method described here using quasi-dynamical downscaling methods (Gutmann et al., 2016).

25 In some cases, the uncertainty introduced by hydrologic model choice could also be significant (e.g., Mendoza et al., 2015; 2016; Mizukami et al., 2016b). We generated preliminary comparisons of hydrologic projections using two different VIC parameter sets as a part of this work, which suggest that the VIC parameters may not significantly influence extreme flow estimates at the reach scale. However, future work could expand this analysis to more rigorously evaluate projections from different parameter sets and/or hydrologic models.

30 In addition to potential issues with the downscaled hydrology, our method is limited by available data on assets exposed to inland flooding. Because the 1% AEP (100 year return interval) floodplains are the only flood risk zones consistently mapped at a national scale, our model tabulates damages only within these mapped floodplains. Our simulations generate damage

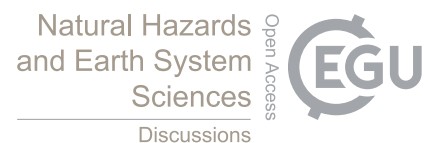

estimates in the early part of the 21st century that are remarkably similar to historical inland flooding damages nationwide (Figure 8), suggesting that our damage estimates are reasonable. However, it is important to stress that we are unable to project damages outside of these mapped floodplains, including flash floods that could be driven by localized extreme precipitation events (e.g., Prein et al., 2016).

Our method also relies on the simplifying assumption that damages from *any* flow exceeding the 1% AEP event can be estimated based on the asset inventory and depth-damage functions tabulated within mapped floodplains. In reality, larger floods will cause more damages than smaller floods, so the magnitude of the flow above the 100-year baseline event will play a role in determining total damages. However, with the exception of limited mapping of 500-year floodplains, there is no national data available to evaluate how damages might increase with increasing flood magnitude above the 1% AEP event.

Similarly, more frequent floods (e.g., the 10- or 50-year event) will still result in monetary costs not evaluated here. Thus our estimates of damages within 100-year floodplains will be minimum estimates for both the baseline and the future time periods. To the extent that floods become both larger and more frequent through time, the degree to which we underestimate flood damages should also increase in the future.

Finally, because there is no *a priori* way to predict how humans will adapt to future flood risk, our model does not account for

current or future adaptations to protect against changing flood frequencies. For similar reasons, we also do not account for population growth or increasing development within flood risk zones. Existing flood control structures are in many cases able to mitigate downstream impacts of extreme flows, such that future changes in the frequency of those flows may not translate directly into increased damages in flood-protected locations. Future demographic and infrastructure changes could also either increase or decrease damages from flooding in the future: increased flood protection measures could decrease damages, while

increases in development in the floodplain could increase them. While it seems clear that exposure to flooding will increase through the 21[st] century in many parts of the United States, the overall damages incurred will depend in large part on how humans adapt to this increasing flood risk.

**Acknowledgements**

The authors acknowledge financial support from the U.S. Environmental Protection Agency's Office of Atmospheric Programs

(contract # EP-BPA-12-H-0024) and the U.S. Army Corps of Engineers (contract # W912HQ-10-D-0005). The views expressed in this article are solely those of the authors, and do not necessary represent those of their employers. The National Center for Atmospheric Research is sponsored by the National Science Foundation.

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

**Figure 1: Locations of the 57,116 stream segment with hydrologic projections used in our analysis. Color corresponds to the baseline average annual maximum flow for each segment, in m³s⁻¹.**





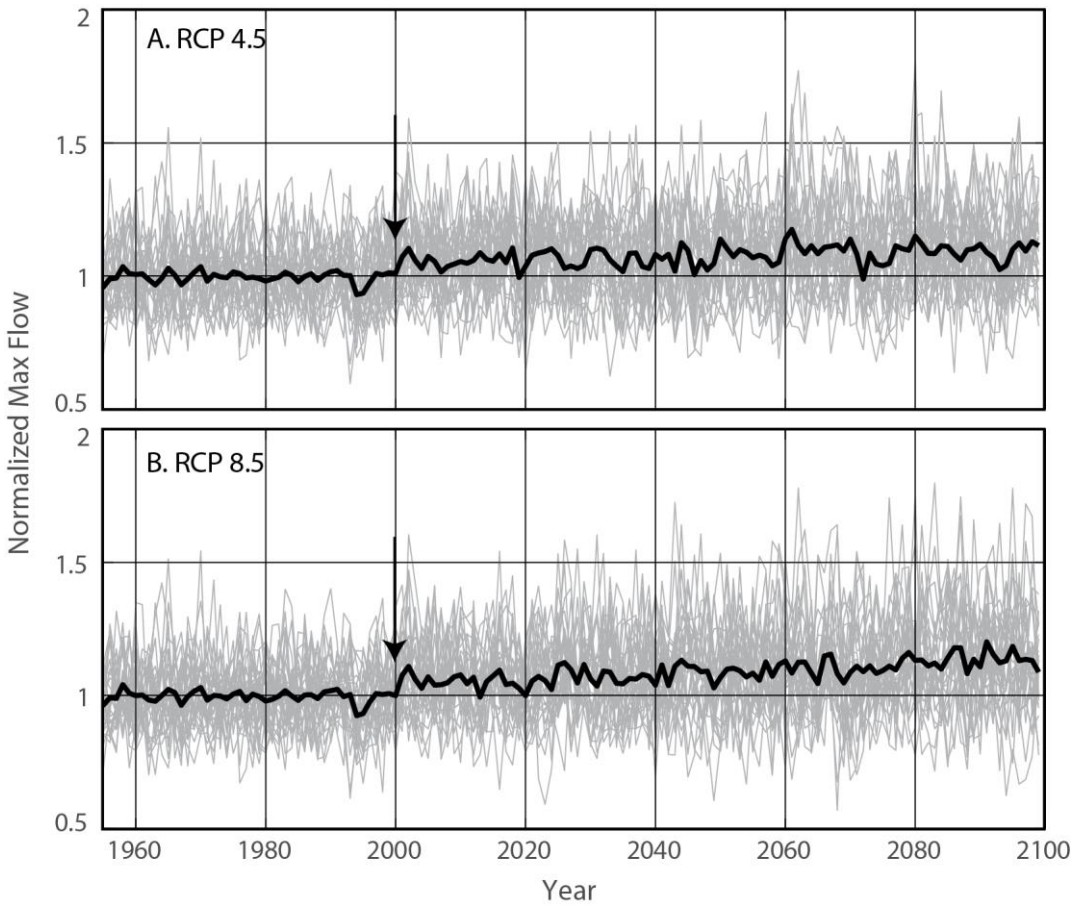

**Figure 2: Trends in annual maximum flow across all stream segments in the CONUS. Thin grey lines represent annual maximum flow normalized to 2001–2020 mean, and averaged across all segments for each individual model. Thick black line represents ensemble average. Step increase in the year 2000 for both RCPs (black arrow) is an artifact of the BCSD method. Accordingly, the "baseline" 1% AEP event was calculated from an early 21$^{st}$ century ensemble (see text).**



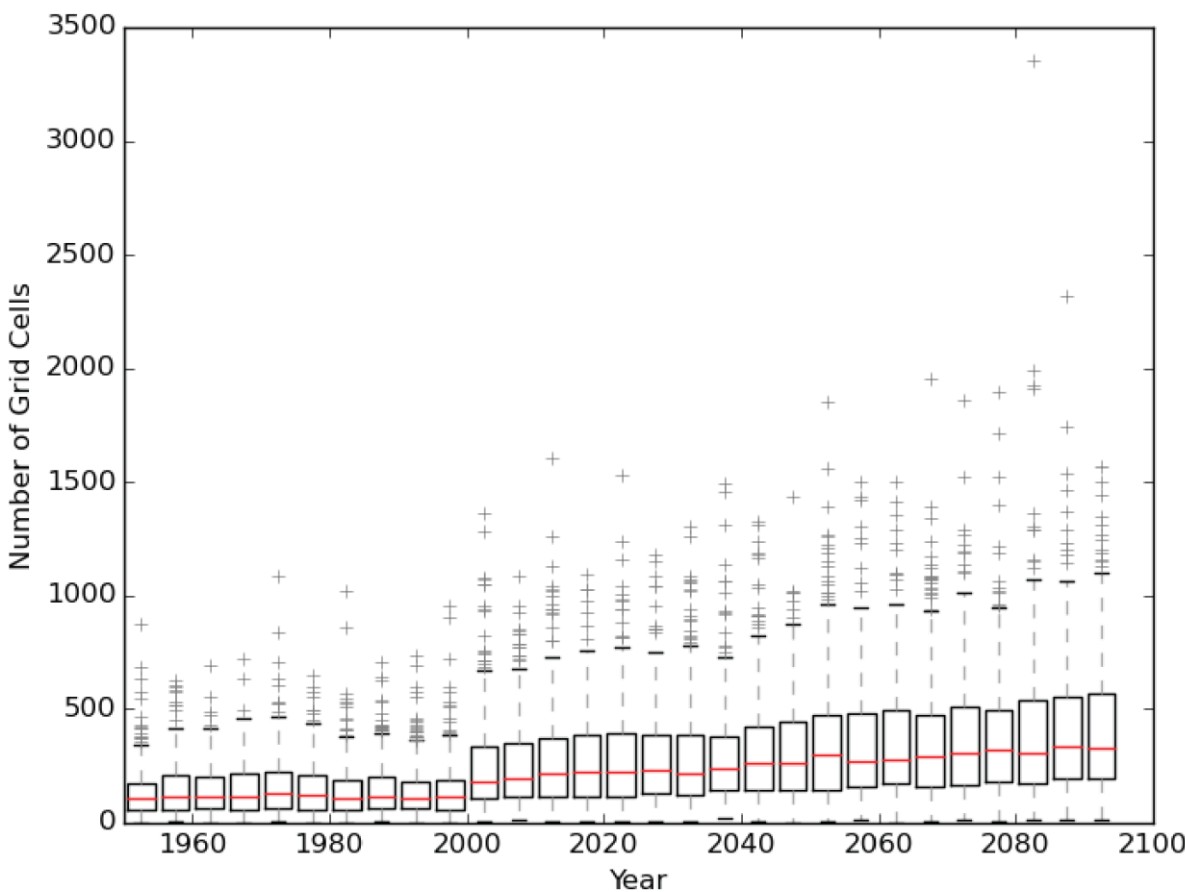

**Figure 3: Number of BCSD grid cells in CONUS experiencing their maximum daily precipitation in each year between 1950 and 2100 per model. Box and whiskers represent spread across all of the individual models used in the flow simulations. The step in the year 2000 (black arrow) is an artifact of the BCSD method (t-test p value <0.00000001). Light grey shading shows period used to calculate "baseline" 1% AEP event.**



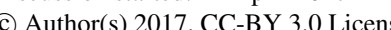


**Figure 4: Total expected damages from a 1% AEP flood event in each of the HRUs in the CONUS. Values in 2014 dollars.**




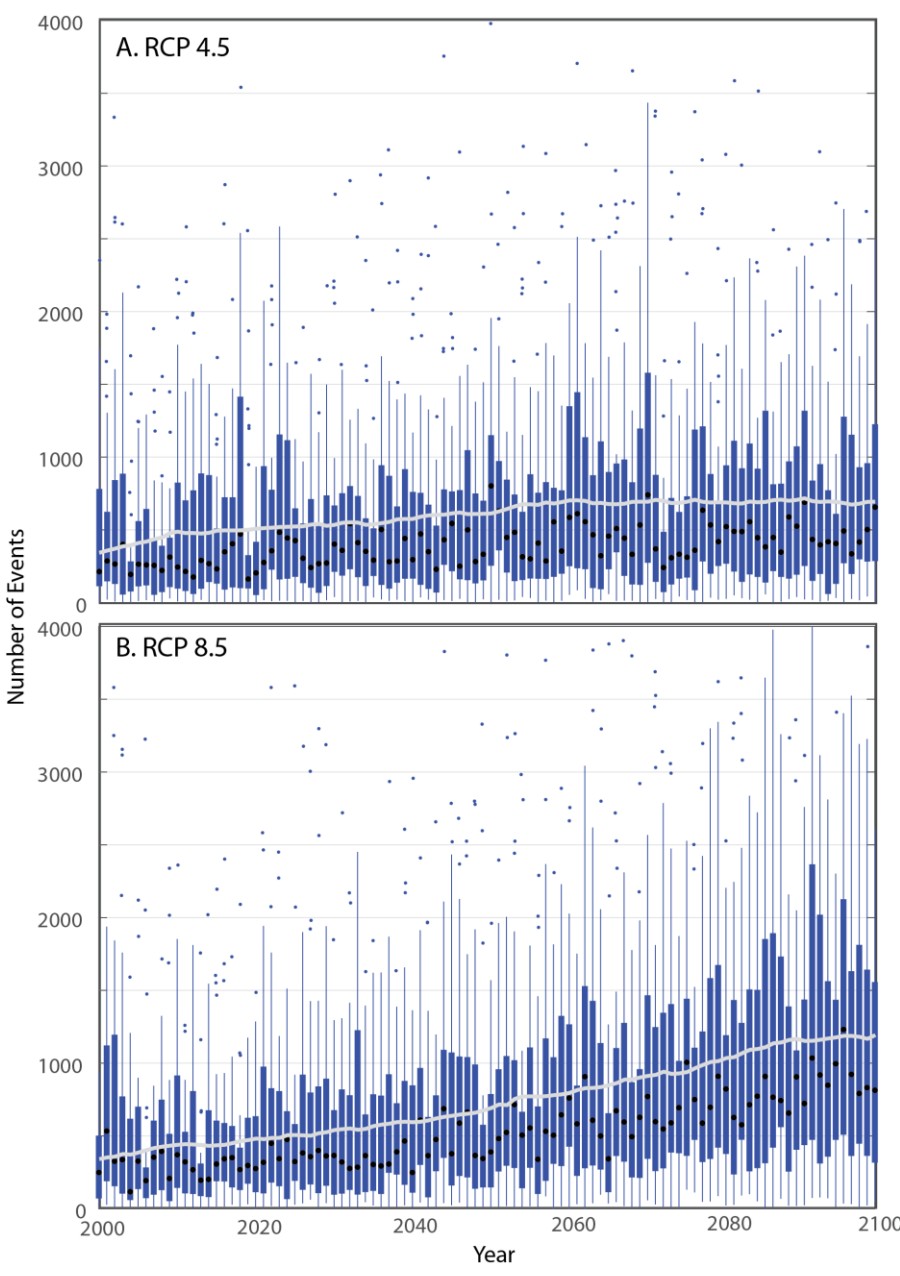

**Figure 5: Number of floods throughout CONUS in each year of the 21st century, across all 29 GCMs in A) RCP4.5 and B) RCP8.5. In each plot, black dots are the median value across all 29 GCMs, thick blue bars are the middle 50% of models, whiskers extend to the 95th percentile of values, and dots represent outliers. The thick grey line is the five-year moving mean across all models. Light grey shading in background shows period used to calculate "baseline" 1% AEP event.**





**Figure 6: Change in frequency of historical 1% AEP events based on ensemble averages for specified RCP and time periods. Calculations are based on individual stream segments over 20-year periods centered on A) 2050 for RCP4.5, B) 2050 for RCP8.5, C) 2090 for RCP4.5, and D) 2090 for RCP8.5. Values are expressed as ratios (e.g., a value of 2 corresponds to a doubling in frequency of the historical 1% AEP event).**




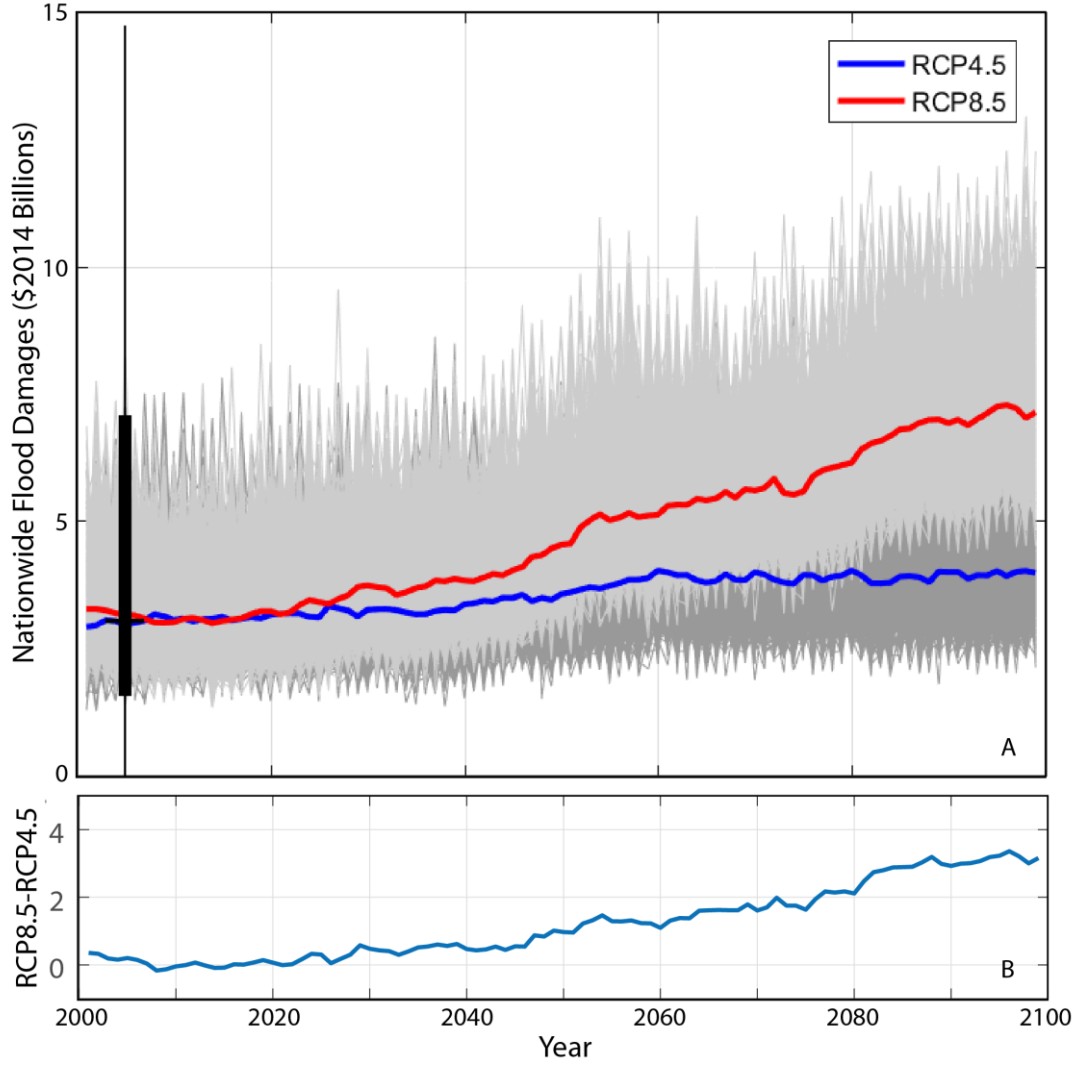

**Figure 7: A) Projected national flood damages within 100-year flood zones, in 2014 dollars. Thin grey lines are 1,000 simulations of damages for RCP4.5 (dark grey) and RCP8.5 (light grey). Blue and red lines are means of simulations for the two RCPs. Box and whisker plot at left is the range of historical observed flooding in CONUS between 1903 and 2014 (10 outliers not shown). B) Difference between mean annual flood damages between RCP4.5 and RCP8.5 (billions of 2014 US dollars).**




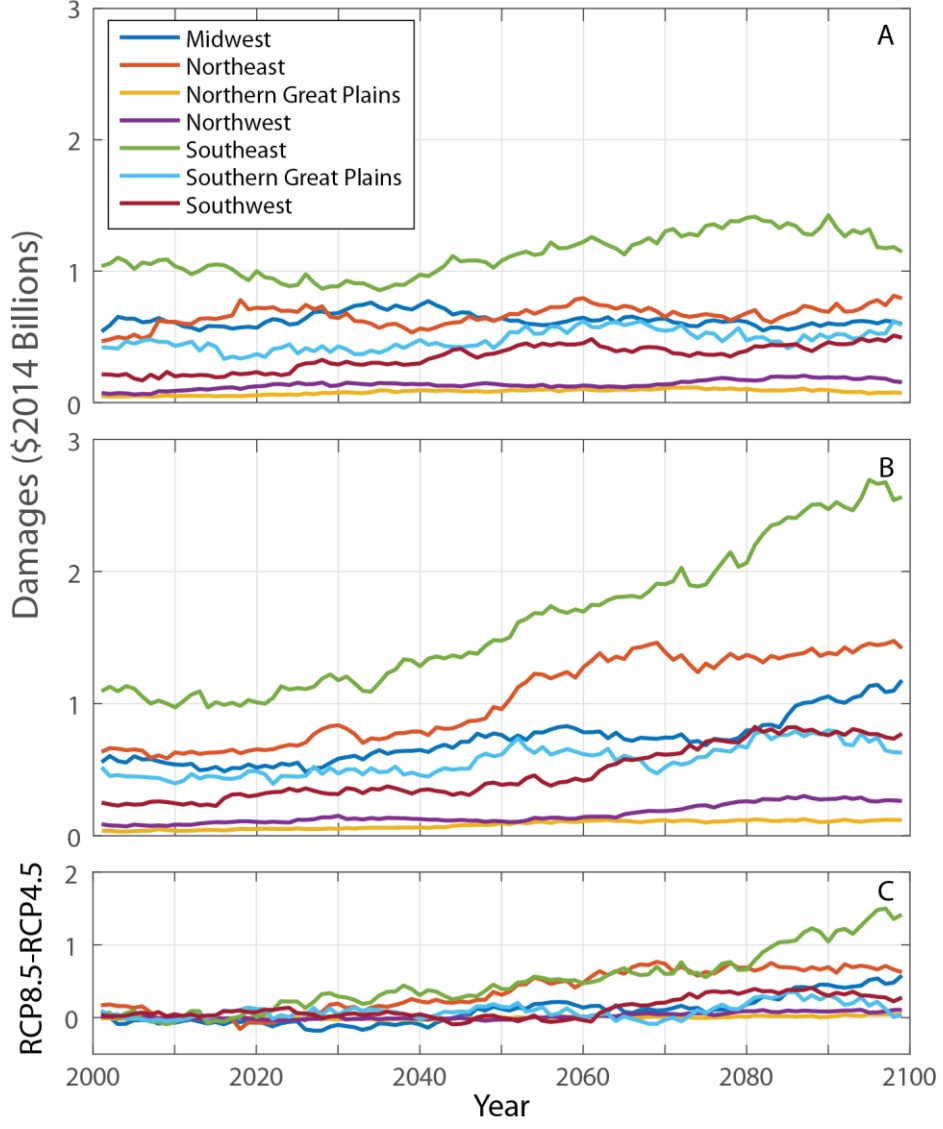

**Figure 8: Average annual flooding damages by region for A) RCP4.5 and B) RCP8.5. C) Difference in annual flood damages between RCP4.5 and RCP8.5 by region (billions of 2014 US dollars).**

