# Peer review of "Climate Change Impacts on 100 year Flood Risk and Asset Damages within Mapped Floodplains of the Contiguous United States"

_Natural Hazards and Earth System Sciences, 2017_

## Referee Comment (RC1) · Anonymous Referee #1 · 22 May 2017

**General comments**

This paper considers how flood frequency and the associated flood damages might evolve depending on different greenhouse gas (GHG) emissions pathways. To my knowledge this is the first paper that proposes an automated methodology at the continental scale to estimate the potential cost of GHG emissions through their effect on flood damages. The manuscript addresses the issue of climate change in monetary terms (the cost of modeled flood damages given different emissions pathways) and as such is a timely and important contribution to the literature, and one which I expect will be of interest to a broad audience. Additionally, the authors frame the issue in a positive manner, by showing how global GHG reductions can be used to limit possible

increases in flood damages.

The methods are mostly well explained. The authors begin with the mapped 1% annual exceedance probability flood extent (100-year flood) across the continental United States. They then assess how these damages will evolve under two different GHG emissions scenarios. Some aspects could be slightly better clarified (see specific comments below), but the authors are generally upfront about the limitations of the work: they state that the projections "should be considered order-of-magnitude estimates..." (p3L.10); they discuss the limitations of GCMs in resolving precipitation (p8), the uncertainties in the hydrological forecasts, the limited data on assets exposed to flooding, the fact that the approach does not account for the effects of changes more/less extreme floods than the 100-year flood, and that it does not take into consideration societal adaptation to flooding. However, these limitations and assumptions are discussed mainly in the conclusions, so the reader is left wondering about some of these matters (e.g. the potential influence of changes in land use) throughout most of the paper. I feel it would help to include a brief statement earlier in the paper, mentioning that the method assumes that there are no changes in land use (no additional constructions in the floodplains, no change in land cover, etc.) and therefore that the overall changes in flood hazard/damage are based mostly on climatic changes.

The overall approach makes sense given the continental scale of the analysis: the authors consider the distribution of results across all 29 GCMs for each RCP and compute the total number of flood events across the CONUS in each year of the model simulation. While this provides an interesting first estimate of potential future changes in flood damages at the continental scale, the uncertainties may be more problematic at the local scale. Also, as stated by the authors, this general approach is relatively conservative and thus likely underestimates the influence of potential increases in extreme precipitation.

In terms of results, I feel that the paper would benefit from a little more explanation. For instance, it is interesting that some regions (like the Southeast) are more affected

by increasing flood damages under RCP8.5 than others, but there is no explanation or suggestion why.

In sum, the paper is very well written, agreeable to read, and aptly illustrated. The technical language is appropriate, and the references are appropriate and accessible. The title and abstract are both pertinent and clear, with an appropriate and complete summary of the contents of the paper.

Specific comments

P2 L.1-5. I feel that this paragraph (on climate attribution) does not fit in very well here – the narrative could be strengthened and clarified.

P2 L.9. Perhaps the authors could state explicitly why those two RCPs were chosen?

P2 L.18. I think there are more recent studies on streamflow trends at the scale of the entire CONUS.

P2 L.22. "Because available hydrologic records tend to be short...". I feel this sentence misses the main point of the paragraph. It seems the issue here is not that historical trends are inconclusive or that existing data records are too short (the USGS database has thousands of sites with more than 50 years of streamflow data; and existing analyses are not all inconclusive), but rather that historical trend analyses are unable to tell us much about the future, and therefore there is increasing interest in using climate model outputs to evaluate future flood risk.

P4 L.21. It's not entirely clear to me how realistic the simulated time series are compared to observed time series- perhaps I missed something; could this be clarified?

P3 L.5. (& discussion P8 L.31) "only the 100-year floodplains are consistently mapped and available at a national scale": for future work, it might be interesting to use an automated digital elevation model floodplain extraction method.

P4 L.3. "Full details of the ... methodologies are available in Mizukami et al. (In Re-

СЗ

view)" – it is difficult to comment on a methodology that is under review in WRR...could the authors comment on this?

P5 L.29. "We created a random sample of flood depths". This section and the calculation of depth-damage function is interesting, but it is a little unclear how the depths were calculated. I assume the bathymetry of the river and any changes in river capacity are not considered; if so, this would be worth commenting on (and the potential implications for the results).

P7 L.17-21. "changes in flood damages broadly mimic changes in flood frequency...". I believe this finding is to be expected, if the method assumes that flood frequency is driven solely by meteorological change, without considering potential temporal changes in the spatial distribution of assets, land use, water management, and/or channel capacity. At this point it would be worth mentioning these assumptions explicitly, rather than waiting until the last paragraph of the manuscript.

P8 L.1. It seems that the difference in projected flood damages between the Southeast and Northeast is considerable (\$2 billion per year by 2100 versus \$1 billion per year by 2100), and would be worth explaining.

P8 L.26. "We generated preliminary comparisons of hydrologic projections using two different VIC parameter sets". This is a little vague and is not explained in the paper; perhaps the authors could be more explicit, or include details in the supplementary materials.

---

## Referee Comment (RC2) · S. Dixon (Referee) · 23 May 2017

This is an interesting paper, combining several modelling approaches to give order of magnitude estimates of economic losses related to 1% flood events increasing with climate change over the 21st century. I enjoyed reviewing it and broadly speaking I think the paper can be published with minor revisions, principally around tightening up some of the language to convey precise meanings. In short I'd recommend the methods are fine as they are, but the discussions need to take extreme care around how far the results can be extrapolated. This is especially important given that the results could have wider public, policy and media interest, and from that perspective it is perhaps even more important to make sure someone reading the paper without all specialist knowledge/training will not potentially misinterpret some of the findings/discussion. Specifically, I think the fact the study is delivering order of magnitude estimates and should be considered a 'first pass' at answering the question of future flood hydrology/risk and damage need to be incorporated into the discussion a little more. Even more importantly this needs to be covered in the abstract for the reasons above.

I should add the caveat that I do not consider myself competent to review all technical aspects of downscaling of GCMs and so would defer to the other reviewers and editor on those aspects of the paper.

Detailed comments: Abstract

10 – The two clauses in the opening sentence don't directly follow from each other. The first part makes link between flood occurrence and extreme weather and says extreme weather events will increase. The second part says therefore flood DAMAGE will increase. Not directly supportable to link increased frequency with increased damage in a general sense. This would need to be amended (at the least) to say "thus [potentially] increasing flood damage.." Or alternatively use a more general concept like increasing risk or exposure.

13 – (and elsewhere – pg 3, line 3). I'm not sure about the terminology of referring to them as "locations", would "reaches" or "catchments" convey this better?

19 – Care in language needed here (and elsewhere). Paper is specifically talking about flood damage, but here talks about flood risk. Not same thing. Would be better to be consistent throughout to avoid confusion.

22 – This sentence needs rewording and maybe more caveats adding. At the moment the argument is somewhat tautological when it's boiled down – "we think we are being conservative, therefore our conclusions are conservative". I think this needs be stated in a way which does not seem to infer what the findings of future work would be! A key issue is that the result is an order of magnitude estimate; there are many assumptions made in the methods (either in choices or models) which are assumed to give uncertainty of an order of magnitude less (hence order of magnitude estimate), but for many of these we don't know whether they are over or under. I think what you are trying to say here is that more advanced techniques can constrain this uncertainty for future work. It's almost a separate point to say that you feel you've made methodological choices which would tend towards underestimating total damage. Indeed it may be worth separating out these two ideas/statements.

Intro 26 – I don't follow this statement I'm confused how an annual average can have a range, or how annual damage can be an average? – I.E. if annual damage is averaged over 100 years it is a single number? Does this mean just the measured annual damage ranges between x and y, or is it estimated from different sources? Or perhaps decadal/regional averages? Clarify.

28 – clarify the "damage" here; is this estimated economic costs, actual rebuild costs, including all economic losses not just physical ones? Important as this relates directly to paper findings so important to know.

29 – Care with language. This flooding is "historical" in what context? Largest ever? Or do you just mean "large flood events"?!

PG2. 1-7 – I think this paragraph could be framed better. I recommend rewording slightly as the three sentences don't seem to exactly follow on, one from the other. In the first it says challenging to understand events to climate change. Then says this is advancing, as well as attributing extremes in general to warming. Then finally says long term trend forecasting is important for stakeholders. At this point you are first making the case for why you would do this work, so I think it would be more powerful to suggest why the approach in the first two sentences is not fit for purpose and so therefore why the trend approach used later on is better/necessary/more useful in an explicit sense. Would be an early marker as to why this is all important and sell it to the reader(s).

12 – be explicit here whether you are talking about the mean damage in the 1% event

per year, or the cumulative damage of all such events over the time span.

15 – I'm uncomfortable with the paper claiming a "deep body of previous work" but not citing any! Is there at least 2-3 review papers that could be cited in terms of "(see A et al, 2006, B & C, 2010….)"

22 – reference(s) for inconclusive studies needed.

23 – references for significant interest, or be more explicit about the source of this if not based on literature.

30 – Use of "flood risk" here, but this time to apply to (I think) the frequency of flood events. If so this is more broadly how I would understand the term, but clashes with usage elsewhere. This needs to be more explicit in this context, or alternatively could define flood risk as a term for purposes of this paper.

PG4. 5 – This needs to be less definitive I think – "are likely to be conservative" rather than "are conservative", unless this is supported with methodological references.

PG5. 20 – reference to the tool needed – ideally to some form of report/paper/website. And also the name of the tool needed.

PG7 23 – this is an interesting use of the word "modest" to refer to $1bn! I take the point, but recommend changing.

PG8 10 – Not sure about "calculate" here, think "estimate" or something similar is more accurate.

PG9 5 – I'm not convinced by the way this is framed. I agree that larger floods can be more damaging, but not necessarily that they always ARE. Likewise, small, more frequent floods can also cause damage, but not always. This will be very catchment and site specific and depend on the floodplain topography and siting of assets. In some cases, it may be that the 1% event floods all assets in a location, and therefore a bigger flood makes no additional difference. I'm therefore a bit uncomfortable with the

certainty that all the estimates are underestimates of damage, particularly given levels of uncertainty in the methods anyway. I'd recommend this section is reworded to be less explicit in predicting the results of refining the methods! Perhaps just highlighting the absence of the frequent small floods and the potential effects of larger events in some (most? many?) catchments and saying it will invariably effect the damage estimates, rather than specifically state your estimates are definitely underestimates of damage in all cases.

Figure 3 – I'm not sure about the p-value reported in the caption. The purpose of a p-value is only to show that it is less than the alpha value set for significance, which is normally 5% or 1% in natural sci. The value of <0.00000001 reported is unnecessary as it doesn't give any more info than something like p<0.001 (0.1%) and may incorrectly imply an incredibly high level of significance is being looked for (as alpha is not explicated stated elsewhere)

Figure 8 – I am perhaps admitting my ignorance of US geography here! But I was not able to easily visualise what the different labelled regions coincided with, particularly given it is being published in a European based journal (albeit an international one) it may be worth adding a map of where you divide up the regions, perhaps this could be incorporated into one of the existing map figures as a background layer to save adding another figure?

Addendum: After typing my report I read the other review comment and noted they have recommended a little more discussion of some of the regional based results. In light of that I really think a map reference of some kind to guide the reader through, as suggested in my figure 8 comment above, would be very helpful.

Simon Dixon University of Birmingham

---

## Author Response (AR1)

August 10, 2017

Paolo Tarolli
Editorial Board
Natural Hazards and Earth System Sciences

Dear Dr. Tarolli:

Thank you for handling our manuscript for Natural Hazards and Earth System Sciences, originally entitled "Modeled changes in 100-year flood risk and asset damages within mapped floodplains of the contiguous United States." Per your request, we have completed a point by point response to the reviewers in this letter, and we have attached a marked-up version of our manuscript documenting the changes we made to our original submission. In addition to this point by point response, we would like to highlight the significant changes we made to the discussion of uncertainties in our paper, as requested. In particular, we added a substantially expanded discussion of all uncertainties in the Methods section of the paper, which now appears in Section 2.4. We moved this discussion of uncertainties up in the paper in response to the concerns raised by the reviewers that the reader needs a better grasp of these uncertainties prior to our discussion of results.

We believe that the changes we have made in response to your comments and those of the reviewers have resulted in a substantially improved manuscript, which we believe will be both acceptable for publication in NHESS, and a significant contribution to this area of natural hazards research.

Our responses to the reviewers' general and specific comments on the manuscript are listed below. In all cases, reviewer comments are marked with an "RC1" or "RC2" denoting which reviewer they are addressing; and our responses are marked with an "AC", reflecting author comments. A marked-up version of the manuscript is attached to end of this letter.

We thank you again for your consideration.
Sincerely,

Cameron W. Wobus, PhD
Senior Scientist
Environment & Health Division

Enc.

*Reviewer #1 General comments*

RC1: This paper considers how flood frequency and the associated flood damages might evolve depending on different greenhouse gas (GHG) emissions pathways. To my knowledge this is the first paper that proposes an automated methodology at the continental scale to estimate the potential cost of GHG emissions through their effect on flood damages. The manuscript addresses the issue of climate change in monetary terms (the cost of modeled flood damages given different emissions pathways) and as such is a timely and important contribution to the literature, and one which I expect will be of interest to a broad audience. Additionally, the authors frame the issue in a positive manner, by showing how global GHG reductions can be used to limit possible increases in flood damages.

The methods are mostly well explained. The authors begin with the mapped 1% annual exceedance probability flood extent (100-year flood) across the continental United States. They then assess how these damages will evolve under two different GHG emissions scenarios. Some aspects could be slightly better clarified (see specific comments below), but the authors are generally upfront about the limitations of the work: they state that the projections "should be considered order-of-magnitude estimates: : :" (p3L.10); they discuss the limitations of GCMs in resolving precipitation (p8), the uncertainties in the hydrological forecasts, the limited data on assets exposed to flooding, the fact that the approach does not account for the effects of changes more/less extreme floods than the 100-year flood, and that it does not take into consideration societal adaptation to flooding. However, these limitations and assumptions are discussed mainly in the conclusions, so the reader is left wondering about some of these matters (e.g. the potential influence of changes in land use) throughout most of the paper. I feel it would help to include a brief statement earlier in the paper, mentioning that the method assumes that there are no changes in land use (no additional constructions in the floodplains, no change in land cover, etc.) and therefore that the overall changes in flood hazard/damage are based mostly on climatic changes.

*AC: We appreciate the need for additional detail in certain parts of the paper. We included a statement earlier in the paper explicitly noting the assumptions so that the reader does not need to be "left wondering" about some of these matters until the end. In response to reviewer and editor comments, we also added a new section on uncertainties much earlier in the paper to make our assumptions and their uncertainties more explicit. Finally, we critically reviewed our manuscript based on specific comments from the Reviewers, and we made further clarifications to the text as requested and summarized below.*

RC1: The overall approach makes sense given the continental scale of the analysis: the authors consider the distribution of results across all 29 GCMs for each RCP and compute the total number of flood events across the CONUS in each year of the model simulation. While this provides an interesting first estimate of potential future changes in flood damages at the continental scale, the uncertainties may be more problematic at the local scale. Also, as stated by

the authors, this general approach is relatively conservative and thus likely underestimates the influence of potential increases in extreme precipitation.

*AC: we agree and acknowledge that the uncertainties from our method are certainly larger at a local scale. We have revised the text to ensure that all of the uncertainties are explicit in the revised manuscript. We also highlighted the limitations of our study in the discussion section of the manuscript.*

RC1: In terms of results, I feel that the paper would benefit from a little more explanation. For instance, it is interesting that some regions (like the Southeast) are more affected by increasing flood damages under RCP8.5 than others, but there is no explanation or suggestion why.

*AC: There are a range of potential reasons why different regions exhibit more significantly increasing flood damages than others. These include differences in the climate change signal, and differences in the distribution of infrastructure within mapped 1% AEP floodplains. We recognize that the manuscript could benefit from more discussion of these nuances, and we have expanded the explanation of topics such as this in the discussion section of our revised manuscript.*

RC1: In sum, the paper is very well written, agreeable to read, and aptly illustrated. The technical language is appropriate, and the references are appropriate and accessible. The title and abstract are both pertinent and clear, with an appropriate and complete summary of the contents of the paper.

*AC: We thank the reviewer for this positive summary of the paper.*

**Reviewer #1 Specific comments**

RC1: P2 L.1-5. I feel that this paragraph (on climate attribution) does not fit in very well here – the narrative could be strengthened and clarified.

*AC: We expanded and revised this paragraph on revision, to provide a more coherent summary of both recent work in climate attribution and the need for long-term projections of trends in flood frequency and magnitude.*

RC1: P2 L.9. Perhaps the authors could state explicitly why those two RCPs were chosen?

*AC: These two RCPs loosely represent a future with little to no action on GHG mitigation (RCP8.5) and one with relatively concerted efforts to reduce GHG emissions (RCP4.5), and as such provide a good backdrop for evaluating how flood damages could be influenced by a change in emissions. These two RCPs are also being recommended for use in the Fourth National Climate Assessment. We made all of these rationales more explicit in the revised manuscript.*

RC1: P2 L.18. I think there are more recent studies on streamflow trends at the scale of the entire CONUS.

*AC: We agree. While our list of examples cited here was not meant to be exhaustive, we have included more recent references on streamflow trends in the CONUS, including Tamaddun et al. (2016); and Ivancic et al. (2017)*

RC1: P2 L.22. "Because available hydrologic records tend to be short…". I feel this sentence misses the main point of the paragraph. It seems the issue here is not that historical trends are inconclusive or that existing data records are too short (the USGS database has thousands of sites with more than 50 years of streamflow data; and existing analyses are not all inconclusive), but rather that historical trend analyses are unable to tell us much about the future, and therefore there is increasing interest in using climate model outputs to evaluate future flood risk.

*AC: We thank the reviewer for pointing out the confusion from this statement. We revised this paragraph to clarify our meaning.*

RC1: P4 L.21. It's not entirely clear to me how realistic the simulated time series are compared to observed time series- perhaps I missed something; could this be clarified?

*AC: This analysis is included in Mizukami et al. (in review), but as noted in the reviewer's comment below (P4 L.3) we recognize that it is difficult for the reader to evaluate this study since it is not yet published. We developed an additional supplemental information file to more clearly describe the salient results from Mizukami et al. to illustrate the degree of correspondence between the simulated and observed timeseries. We also note that the results from our study are based on a delta approach – that is, we are not using absolute magnitudes of flow to drive any of our modeling results; only changes in frequency of events exceeding a model-derived threshold. We included additional discussion in the body of the manuscript that summarizes all of these points.*

RC1: P3 L.5. (& discussion P8 L.31) "only the 100-year floodplains are consistently mapped and available at a national scale": for future work, it might be interesting to use an automated digital elevation model floodplain extraction method.

*AC: We agree that it would be interesting and informative to repeat this analysis for a wider range of flood magnitudes. We have done some preliminary analysis of assets exposed to a wider range of flood magnitudes using data from the FEMA Risk MAP program, and we have included some discussion of this as a worthwhile avenue for future research in our revision. However, extending our results using an automated DEM extraction method would be a significant undertaking, and one that is well beyond the scope of this study.*

RC1: P4 L.3. "Full details of the…methodologies are available in Mizukami et al. (In Review)" – it is difficult to comment on a methodology that is under review in WRR…could the authors comment on this?

*AC: Mizukami et al. (in review) has been revised and resubmitted in response to reviewer comments, and we anticipate that the manuscript should be in press relatively soon. However, as noted above we have included salient details of the Mizukami et al. paper in our supplemental information file to ensure that the reader has enough information available to understand the method.*

RC1: P5 L.29. "We created a random sample of flood depths". This section and the calculation of depth-damage function is interesting, but it is a little unclear how the depths were calculated. I assume the bathymetry of the river and any changes in river capacity are not considered; if so, this would be worth commenting on (and the potential implications for the results).

*AC: We clarified this part of the description in revision. In addition, we have added a reference to a presentation on the National Flood Risk Characterization tool in the manuscript. This reference contains many of the details sought by the reviewer here.*

RC1: P7 L.17-21. "changes in flood damages broadly mimic changes in flood frequency…". I believe this finding is to be expected, if the method assumes that flood frequency is driven solely by meteorological change, without considering potential temporal changes in the spatial distribution of assets, land use, water management, and/or channel capacity. At this point it would be worth mentioning these assumptions explicitly, rather than waiting until the last paragraph of the manuscript.

*AC: We thank the reviewer for pointing this out. Our revision is more explicit about assumptions at this point in the manuscript, as well as in the discussion. In particular, we have reiterated that the lack of modeled changes in land use and infrastructure in our method mean that changes in flood frequency are driven entirely by changes in climate forcing.*

RC1: P8 L.1. It seems that the difference in projected flood damages between the Southeast and Northeast is considerable ($2 billion per year by 2100 versus $1 billion per year by 2100), and would be worth explaining.

*AC: Given that our results are primarily driven by changes in precipitation, this result indicates that precipitation and runoff during the months that cause flooding are projected to increase more consistently in the Southeast than in the Northeast, and/or that there is more infrastructure at risk in the Southeast than in the Northeast. We have expanded the discussion to more explicitly address this finding and our hypotheses for why this is occurring; however, this is in large part an avenue for future research.*

RC1: P8 L.26. "We generated preliminary comparisons of hydrologic projections using two different VIC parameter sets". This is a little vague and is not explained in the paper; perhaps the authors could be more explicit, or include details in the supplementary materials.

*AC: We expanded this discussion, and we have also included more information on this topic in the new supplemental information file we developed for our resubmittal.*

**Reviewer #2 General Comments**
RC2: This is an interesting paper, combining several modelling approaches to give order of magnitude estimates of economic losses related to 1% flood events increasing with climate change over the 21st century. I enjoyed reviewing it and broadly speaking I think the paper can be published with minor revisions, principally around tightening up some of the language to convey precise meanings. In short I'd recommend the methods are fine as they are, but the discussions need to take extreme care around how far the results can be extrapolated. This is especially important given that the results could have wider public, policy and media interest, and from that perspective it is perhaps even more important to make sure someone reading the paper without all specialist knowledge/training will not potentially misinterpret some of the findings/discussion. Specifically, I think the fact the study is delivering order of magnitude estimates and should be considered a 'first pass' at answering the question of future flood hydrology/ risk and damage need to be incorporated into the discussion a little more. Even more importantly this needs to be covered in the abstract for the reasons above.

I should add the caveat that I do not consider myself competent to review all technical aspects of downscaling of GCMs and so would defer to the other reviewers and editor on those aspects of the paper.

*AC: We thank the reviewer for this overall positive review. As with the comments from Reviewer #1, we recognize the need for additional clarity/explanation of some components of the paper, and have addressed these concerns on revision.*

*Reviewer #2 Specific Comments*

RC2: 10 – The two clauses in the opening sentence don't directly follow from each other. The first part makes link between flood occurrence and extreme weather and says extreme weather events will increase. The second part says therefore flood DAMAGE will increase. Not directly supportable to link increased frequency with increased damage in a general sense. This would need to be amended (at the least) to say "thus [potentially] increasing flood damage.." Or alternatively use a more general concept like increasing risk or exposure.

*AC: We recognize the potential incongruity as written. We have revised the text as suggested.*

RC2: 13 – (and elsewhere – pg 3, line 3). I'm not sure about the terminology of referring to them as "locations", would "reaches" or "catchments" convey this better?

*AC: Agreed. We have revised "locations" to "reaches" here and elsewhere for consistency and clarity.*

RC2: 19 – Care in language needed here (and elsewhere). Paper is specifically talking about flood damage, but here talks about flood risk. Not same thing. Would be better to be consistent throughout to avoid confusion.

*AC: We have clarified the use of "damages" vs "risk" throughout the manuscript.*

RC2: 22 – This sentence needs rewording and maybe more caveats adding. At the moment the argument is somewhat tautological when it's boiled down – "we think we are being conservative, therefore our conclusions are conservative". I think this needs be stated in a way which does not seem to infer what the findings of future work would be! A key issue is that the result is an order of magnitude estimate; there are many assumptions made in the methods (either in choices or models) which are assumed to give uncertainty of an order of magnitude less (hence order of magnitude estimate), but for many of these we don't know whether they are over or under. I think what you are trying to say here is that more advanced techniques can constrain this uncertainty for future work. It's almost a separate point to say that you feel you've made methodological choices which would tend towards underestimating total damage. Indeed it may be worth separating out these two ideas/statements.

*AC: We appreciate the reviewer's concern about our use of the word "conservative" here. The reviewer is correct that the point of this sentence is not to imply that we are underestimating or overestimating damages, but to state that further refinements to our methods could improve our understanding of results and their sensitivity to methodological choices. We have clarified the text accordingly.*

RC2: Intro 26 – I don't follow this statement I'm confused how an annual average can have a range, or how annual damage can be an average? – I.E. if annual damage is averaged over 100 years it is a single number? Does this mean just the measured annual damage ranges between x and y, or is it estimated from different sources? Or perhaps decadal/regional averages? Clarify.

*AC: We recognize the confusion in the text as currently written. The intended meaning here was that nationwide inland flooding damages each year typically fall within a range of dollar values. We have clarified our meaning to ensure that there is no confusion.*

RC2: 28– clarify the "damage" here; is this estimated economic costs, actual rebuild costs, including all economic losses not just physical ones? Important as this relates directly to paper findings so important to know.

*AC: The numbers we quote here are reported damages in terms of physical damage to property, as summarized in NOAA (2016). We have clarified the text as requested.*

RC2: 29 – Care with language. This flooding is "historical" in what context? Largest ever? Or do you just mean "large flood events"?!

*AC: We are referring to these events only as "very large flood events." We have clarified the text accordingly.*

RC2: PG2. 1-7 – I think this paragraph could be framed better. I recommend rewording slightly as the three sentences don't seem to exactly follow on, one from the other. In the first it says challenging to understand events to climate change. Then says this is advancing, as well as attributing extremes in general to warming. Then finally says long term trend forecasting is important for stakeholders. At this point you are first making the case for why you would do this work, so I think it would be more powerful to suggest why the approach in the first two sentences is not fit for purpose and so therefore why the trend approach used later on is better/necessary/more useful in an explicit sense. Would be an early marker as to why this is all important and sell it to the reader(s).

*AC: We thank the reviewer for suggesting ways we could improve this paragraph. We have revised the paragraph to improve the way this part of the introduction frames the remainder of the manuscript.*

RC2: 12 – be explicit here whether you are talking about the mean damage in the 1% event per year, or the cumulative damage of all such events over the time span.

*AC: We have clarified the text in this part of the manuscript, to make it explicit that our study looks at projected damages from 1% events in each year, based on an ensemble across a suite of GCMs.*

RC2: 15 – I'm uncomfortable with the paper claiming a "deep body of previous work" but not citing any! Is there at least 2-3 review papers that could be cited in terms of "(see A et al, 2006, B & C, 2010: : :.)"

*AC: We thank the reviewer for pointing this out. As written, we reference some of the "deep body of previous work" in the sentences following this one. However, we have revised to bring some of these references up further in the paragraph to avoid the incongruity pointed out by the reviewer.*

RC2: 22 – reference(s) for inconclusive studies needed.
RC2: 23 – references for significant interest, or be more explicit about the source of this if not based on literature.

*AC: We have revised this paragraph and the paragraph preceding it to make better reference to the previous literature in the context of our own study.*

RC2: 30 – Use of "flood risk" here, but this time to apply to (I think) the frequency of flood events. If so this is more broadly how I would understand the term, but clashes with usage elsewhere. This needs to be more explicit in this context, or alternatively could define flood risk as a term for purposes of this paper.

*AC*: *We thank the reviewer for pointing out our use of this potentially confusing term. We have clarified our intended meaning of "flood risk" throughout the paper*

RC2: PG4. 5 – This needs to be less definitive I think – "are likely to be conservative" rather than "are conservative", unless this is supported with methodological references.

*AC*: *We have clarified the text as requested.*

RC2: PG5. 20 – reference to the tool needed – ideally to some form of report/paper/website. And also the name of the tool needed.

*AC*: *We have included reference to a publicly available presentation so that the flood risk tool is more fully referenced. This presentation is available here:* http://www.iwr.usace.army.mil/Portals/70/docs/frmp/Flood_Risk_Char/NFRCT_Slides_FRM_wkshp_v1.pdf

RC2: PG7 23 – this is an interesting use of the word "modest" to refer to $1bn! I take the point, but recommend changing.

*AC*: *We thank the reviewer for pointing this out. We have revised the language here.*

RC2: PG8 10 – Not sure about "calculate" here, think "estimate" or something similar is more accurate.

*AC*: *We have revised to use "estimate" instead of "calculate".*

RC2: PG9 5 – I'm not convinced by the way this is framed. I agree that larger floods can be more damaging, but not necessarily that they always ARE. Likewise, small, more frequent floods can also cause damage, but not always. This will be very catchment and site specific and depend on the floodplain topography and siting of assets. In some cases, it may be that the 1% event floods all assets in a location, and therefore a bigger flood makes no additional difference. I'm therefore a bit uncomfortable with the certainty that all the estimates are underestimates of damage, particularly given levels of uncertainty in the methods anyway. I'd recommend this section is reworded to be less explicit in predicting the results of refining the methods! Perhaps just highlighting the absence of the frequent small floods and the potential effects of larger events in some (most? many?) catchments and saying it will invariably effect the damage estimates, rather than specifically state your estimates are definitely underestimates of damage in all cases.

*AC*: *We appreciate this comment from the reviewer, and also recognize this to be a clear avenue for future research. We have reworded this section to more clearly reflect what we can and cannot infer from our results.*

RC2: Figure 3 – I'm not sure about the p-value reported in the caption. The purpose of a p-value is only to show that it is less than the alpha value set for significance, which is normally 5% or 1% in natural sci. The value of <0.00000001 reported is unnecessary as it doesn't give any more info than something like p<0.001 (0.1%) and may incorrectly imply an incredibly high level of significance is being looked for (as alpha is not explicated stated elsewhere)

*AC: We appreciate this comment and we have revised the figure caption accordingly.*

RC2: Figure 8 – I am perhaps admitting my ignorance of US geography here! But I was not able to easily visualise what the different labelled regions coincided with, particularly given it is being published in a European based journal (albeit an international one) it may be worth adding a map of where you divide up the regions, perhaps this could be incorporated into one of the existing map figures as a background layer to save adding another figure?

Addendum: After typing my report I read the other review comment and noted they have recommended a little more discussion of some of the regional based results. In light of that I really think a map reference of some kind to guide the reader through, as suggested in my figure 8 comment above, would be very helpful.

*AC: These region labels are included in Figure 4, but we recognize that they could be missed if they are not described more explicitly in figures such as Figure 8. We have added reference to Figure 4 here and elsewhere in the manuscript, so that it is more clear to the reader where each of the regions is located.*

[revised manuscript text omitted]

---

## Author Response (AR2)

October 6, 2017

Paolo Tarolli
Editorial Board
Natural Hazards and Earth System Sciences

Dear Dr. Tarolli:

We are in receipt of your decision letter regarding our manuscript nhess-2017-152. We appreciate your time and each of the Referees' time in providing a careful critique of our methods and our results.

Based on the comments from Referee #2 (Simon Dixon), it appears that we have addressed all of his concerns, with the exception of adding one reference to our list of references. We have added that reference to our revised manuscript, and we thank Dr. Dixon and Anonymous Referee #1 again for their time and their overall positive reviews of our work.

In the detailed response to comments below, and in the attached manuscript revision, we focus on addressing the new comments from Anonymous Referee #3. However, based on our review of the new issues raised by Referee #3, it is clear that this Referee actually did not review the latest version of the manuscript, and instead reviewed our original submission dated 24 April, 2017. As a result, we believe that the majority of the concerns raised by Referee #3 have already been addressed. Below we summarize the detailed comments from Referee #3, along with a summary of changes we either already made in our August revision or that we have updated since our receipt of the decision letter dated earlier this week.

We thank you once again for your consideration.
Sincerely,

Cameron W. Wobus, PhD
Senior Scientist
Environment & Health Division

Enc.

Below we summarize new comments from Referee #2 and Referee #3. We stress that because Referee #3 does not appear to have read our August 10, 2017 revision, we believe that many of his/her comments have already been addressed by the careful revisions we have already made. In our responses to Referee #3 below, we point to those revisions, where appropriate, as well as to additional changes we have made to the 10 August 2017 paper to address other comments.

In all cases, the Referee's comments are in plain type and our responses and commentary are in italics.

**Referee #2 General Comments**

I have reviewed the response of the authors' and the corrected document and I am happy to confirm they have addressed all the comments I raised in my initial review (as well as those of the other reviewer). The authors have clearly put a lot of effort into making improvements to the manuscript and I have no hesitation in recommending the revised paper for publication. I did notice one minor issue in that "Maurer et al, 2002" in the supplemental information does not appear in the reference list - this will need adding, but I leave it to the editor whether this can be done during pdf proofs post-acceptance or will need adding beforehand.

*We thank Dr. Dixon for his thorough reviews of our initial manuscript, for agreeing to serve as a reviewer for this second round of reviews, and for his overall positive view of our work. We have added Maurer et al., 2002 to our list of references.*

**Referee #3 General comments**

The proposed work deals with the examination of future changes in the frequency of floods across contiguous United States. A methodology is proposed to assess future flood impacts in terms of monetary damages to assets. The topic is of broad interest and the methodology described involves novel elements. While the authors admittedly have applied and described a comprehensive framework for addressing the issue of future flood risk and associated impacts, I have a number of concerns that need to be addressed in my opinion to improve the clarity of the work and make this contribution more meaningful.

*We agree with Referee #3 that this topic is of broad interest, and we thank the Referee for recognizing the novelty and comprehensive nature of our approach. We also thank the Referee for the detailed comments, which are addressed below. However, based on Referee #3 comment 2, in which this Referee specifies line numbers of particular passages in the text, it appears that this Referee did not actually review the version of the manuscript wherein we carefully addressed the comments from the first round of reviews (dated 10 August 2017). Instead, Referee #3 appears to have reviewed our initial submittal (dated 24 April, 2017), which has been significantly improved, and which already addresses many of this Referee's concerns. Below we list the two quotes from our text that demonstrate that this Referee reviewed the earlier version of our paper:*

| Quotation and Line number cited by Referee #3 | Line(s) in 24 Apr 2017 Submittal | Line(s) in 10 Aug 2017 Revision |
| --- | --- | --- |
| L4-5, p.5: "...there were no systematic differences in results in the annual maximum time series…" | L4-5, p. 5 | L30-31, p. 4 |
| L23, p5: "we used a series of steps to calculate the depth of flooding" | L23, p.5 | L.15-16, p.5 |

*In the responses to specific comments below, we summarize new changes we have made in direct response to this Referee's comments. Where appropriate, we also summarize the changes that we already made in our 10 August, 2017 revision that directly address this Referee's comments.*

**Specific Comments from Referee #3**

1. First of all the title is somewhat misleading because it is focused on "100yr" flood. In fact the authors investigate frequency of floods exceeding a threshold considered to be the 100yr flood from the early 21st century. I think that removing the "100yr" would make the title more representative of the work.

   *Because our paper explicitly models only events exceeding the 100-year magnitude flood event, we believe that the title accurately reflects what is in the paper. We note that this concern echoes some of the concerns raised in the first round of reviews, which we have addressed in the 10 August, 2017 version of our manuscript. In particular, we added an entire section called "Uncertainties" to our revised manuscript, which specifically addresses the uncertainties associated with our assumption that damages from any flood exceeding the "100-year" event are being approximated by the damages associated with the mapped 100-year floodplain.*

2. A number of important aspects are simply mentioned without providing any evidence that support them. For example:

   a. L15-16,p1: "Our model generates early 21st century flood damages that reasonably approximate….". You need to provide some evidence on this

   *The selected quotation comes from the abstract, which we believe is an inappropriate place to provide details to back this statement up. Figure 7 in the manuscript illustrates the correspondence between modeled early 21$^{st}$ century flood damages and observations; we have also added a sentence to Section 3.2 to more clearly highlight this part of Figure 7.*

1881 Ninth Street   Suite 201   Boulder, CO 80302   Office 303.381.8000   abtassociates.com

b. L4-5,p5: "…there were no systematic differences in results in the annual maximum time series…". How did you evaluate this? Please specify and report associated metrics/results.

*We generated a series of figures like the one below to compare annual maxima across all models for a selected set of nodes. From this qualitative summary, we found that no models were systematically higher or lower than the others in their estimate of the annual maximum flows across all nodes: some models' annual maxima were higher at one node, but lower at another. The point of the paragraph cited by the Referee is to justify our use of a full ensemble to estimate the 1% AEP event. Although we did this comparison across models and can justify this statement, we do not believe this sentence is actually required, particularly given the expanded discussion of uncertainties that we have included in Section 2.4 and documented in Supplemental Information File #2. As a result, we have opted to remove this sentence to avoid confusion.*

[Figure]

c. L23,p5: "we used a series of steps to calculate the depth of flooding." This is an important methodological step. It should be clearly discussed and demonstrated in the manuscript (rather than provided as supplement).

*This comment has already been addressed in our revised manuscript. The earlier version of the manuscript that the Referee read contained significantly less information on how the damage distributions were calculated. The updated version includes a substantially expanded discussion of this in Supplemental Information File #3, and a reference in Section 2.3 to a document (IWR, 2014) which describes all of the details the Referee is seeking.*

3.  A critical element of this work is the identification of the 1% AEP from the 2001-2020 period and its connection to the 100yr flood maps generated by FEMA. To be consistent in your analysis, and be able to connect the 100yr flood map with the 100yr estimate of your simulations you should use the same period that FEMA used to estimate the 100yr flood. In that case you will at least sample the same climate period. Otherwise you may be introducing a systematic error in your analysis. Grouping results from all models does not resolve this, given that all models are realizations of the same climate period. I understand that the issue of "data jump" after year 2000 prevents you from using the before 2000 simulations, however the point of uncertainty in the 100yr flood estimated by the last 20yrs could be demonstrated by observations at selected locations (e.g. USGS stations).

    *We disagree with the assessment of the Referee, for two reasons. First, we note that the core of our methodology is to examine changes in the frequency of **modeled** 1% AEP events. These changes are based on model to model comparisons through the 21ˢᵗ century. While we recognize that the time period of our "baseline" does not coincide with the time period during which the 100-year floodplains were mapped, this is likely to make our results conservative if one assumes that extreme hydrologic events have already been increasing through the late 20ᵗʰ century.*

    *Second, because we are using a delta method, we neither expect nor require our model to perfectly simulate 100-year events in any one location. The mapped 100-year floodplains are used only to calculate the damages associated with 1% AEP events in each stream reach, and our method does not require that we demonstrate perfect correspondence between modeled and observed hydrologic extremes. We believe that the discussion of the model performance in Supplemental Information File #1 and in Mizukami et al., (2017), both of which were added to the revised manuscript, provide sufficient documentation of the model's performance to justify its use under the delta method framework we applied. Furthermore and as described above, Figure 7 demonstrates that despite the uncertainties described above and in Section 2.4, the simulated damages in the baseline model run very closely mimic observed flood damages over the late 20ᵗʰ century.*

4.  A major concern of the validity of the results is that there is no sort of evaluation of the performance of the model used. Can you provide any indication that the VIC-based hydrologic simulations can reproduce a realistic flood response at least for the period 2000-2017, where observations exist? Otherwise in the absence of any evaluation procedure findings are highly questionable.

    *We believe that our 10 August 2017 revision already addresses this comment. We agree that the original version of our manuscript, which this Referee is referencing, did not have sufficient documentation of the VIC-based hydrologic simulations. We added an entire supplemental information file to address this, and the paper summarizing the VIC*

*modeling method and results (Mizukami et al., 2017) is now out and referenced in the paper.*

5. Although the connection of changes in flood frequency with monetary damages is a novel element and generally welcome, I feel that it is the weak link of the study and should be minimized to a section that offers some indication rather than be the central part of the work. The authors themselves in their conclusions acknowledge a number of very important sources of uncertainty and factors that have not been taken into account in the damage estimates. Acknowledging is good but not enough to justify why we should then consider meaningful actual numbers of future flood damages when population expansion for the next 80 yrs is not even considered.

*We believe that this comment has already been addressed in our revised manuscript. We agree that the changes in monetary damages is a novel element, and we further believe that it is actually one of the most important contributions of this paper, and is sufficiently supported with robust methods. The previous Referees asked for additional discussion of uncertainties, which we addressed by adding Section 2.4 to explicitly discuss the full range of uncertainties associated with our method, including changes in population. We have also added reference to the two papers suggested by the Referee, as summarized below.*

6. I would suggest the authors to restructure the work and have as central point of the work the analysis of changes in flood frequency under different climate scenarios. An effort for a more in-depth explanation of the patterns in Fig. 6 would be mostly welcome. The association of predicted increases with potential damages should remain as a secondary element in the paper but analysis should be mostly focused on the relative differences between the two emission scenarios rather than an effort of providing actual numbers. Translation of future flood risk increases into damages is indeed a highly complex topic. I believe that the authors can benefit from a more thorough literature review on this as well. See for example

Liu, Jing, et al. "Future property damage from flooding: sensitivities to economy and climate change." Climatic change132.4 (2015): 741-749.

Bubeck, P., et al. "How reliable are projections of future flood damage?." Natural Hazards and Earth System Sciences 11.12 (2011): 3293.

*See response to comment #5 above. We agree that changes in flood frequency are an important element of this work; however, the link between flood frequency and flood damages is an important and novel aspect of our work, and a part that we do not feel should be de-emphasized. In our response to the initial round of reviews, we increased the discussion of uncertainties related to the translation of flood frequency to damages*

*(see Section 2.4 in the revised manuscript), which we also believe helps to address this Referee's comment.*

*The patterns shown in Figure 6 and in the regional damage results shown in Figure 8 were pointed out by both reviewers in the last round of reviews. In response to those comments, we expanded the discussion of these patterns as part of our last round of revisions. We refer the Referee to our detailed responses to those reviews and the expanded discussion section in the 10 August, 2017 paper.*

[revised manuscript text omitted]

---

## Author Response (AR4)

October 26, 2017

Paolo Tarolli
Editorial Board
Natural Hazards and Earth System Sciences

Dear Dr. Tarolli:

We are in receipt of your most recent decision letter regarding our manuscript nhess-2017-152 requesting minor revisions. This letter details our responses to the latest comments from Reviewer #3. In all cases, the Referee's comments are in plain type and our responses and commentary are in italics. We appreciate your time and the time of all of the Referees in providing a careful critique of our methods and our results.

We thank you once again for your consideration.
Sincerely,

Cameron W. Wobus, PhD
Senior Scientist
Environment & Health Division

Enc.

1881 Ninth Street   Suite 201   Boulder, CO 80302   Office 303.381.8000   abtassociates.com

*Referee #3 General Comments*

I have reviewed the revised manuscript and the author's response. Indeed, I had accidentally reviewed the original version of the manuscript and the revised version is certainly improved. As I stated before, I consider the methodological framework presented in this work novel and as such, I wish to see the paper published.

*We thank the reviewer for the overall positive review. We wish to see the paper published as well.*

*Referee #3 Specific Comments*

1. I am fine with your methodology for assessing changes in modeled 1% AEP events, as you mention. However, your assessment of changes in frequency is based on a threshold (the one you identified at the 1% AEP from the 2001-2020 period), which means that your results are also threshold-dependent. Considering a longer period could (certainly would) lead to a different threshold that would lead to potential differences in the results. In fact, this relates to your discussion on the sampling uncertainty which you have identified to be in the order of 5-20% that you chose not to propagate in your assessment. However, presenting some evidence on the sensitivity of the results on the threshold (even just for the min/max possible values) would be beneficial for the readers. Again, due to the complexity of the problem you are addressing it is understandable that all the different sources of uncertainty cannot be realistically quantified. However, since you have already estimated a range of uncertainty for the 1% AEP I would strongly encourage you to show how this affects the results.

   *We recognize and appreciate the Reviewer's concern, and we have evaluated the effects of this threshold-dependence on the timeseries of flooding, as follows: based on our bootstrapping analysis, we assigned a random error to each 1% AEP flood ranging from +20% to -20%, and re-calculated the timeseries of flooding at each node for each model. We then calculated the average number of floods occurring in the CONUS each year, with and without this error propagation, for each model, and we also calculated the distribution in the number of floods across the full ensemble. We have summarized the results of this exercise in Section 3.1 of the revised manuscript, and we have added two new figures and a summary to Supplemental Information File #2. Because the error on the 1% AEP event can be positive or negative based on our bootstrapping analysis, the overall effect on the timeseries of floods (and therefore damages) nationwide is negligible: while some nodes experience more floods, others experience fewer floods. As we expected, the overall impact is also minor in comparison to the intermodel variability, and justifies our decision to leave this source of uncertainty out of the remainder of the paper.*

2. I understand the point of the authors and my point was not to require that the simulated flows matched perfectly the observed. My point connects to the previous point on the impact of uncertainty in the estimation of 1% AEP. Let's assume that FEMA calculated

the 100yr flood based on N observations available from the simulations. Take the N simulated events and calculate the 100yr flood (Q100N). If we take a sub-sample n<N of those observations and recalculate the 100yr flood we will end up with a different estimate Q100n (uncertainty due to sample size). The number of times the future (simulated) flows exceed Q100N and Q100n will be different and thus the associated estimates of future damages would be different. Again, in my opinion, this point could be addressed (at least partially) by providing some indications on the sensitivity of the results to the estimation of the baseline 1% AEP.

*See response to comment #1 above: we believe that the new error analysis we conducted, as summarized above, addresses this comment as well.*

3. My second comment relates to page 5, L5 where the authors define as "flood" the annual maximum flow value that exceeds the baseline 1% AEP. Why are you considering only the annual max? There can be also other events within the year that exceed the baseline 1% AEP. Accounting for these as well will have a significant impact in the future change of "flood" frequency and associated damages. Please clarify/justify your choice.

*The reviewer is correct that although it is unlikely, more than one event could occur in a given year that would exceed the historical 1% AEP event. However, because our 1% AEP event calculations are based on an annual maximum timeseries, we have focused the remainder of our analysis on the annual maximum flows as well. We have added additional text to Section 2.4 describing this choice, and why we anticipate that this choice makes our future flood damage estimates conservative.*

[revised manuscript text omitted]